# Azathioprine immunosuppression and disease modification in Parkinson's disease (AZA-PD): a randomised double-blind placebo-controlled phase II trial protocol

Julia C Greenland ,[1] Emma Cutting,[1,2] Sonakshi Kadyan,[2] Simon Bond,[2] Anita Chhabra,[3] Caroline H Williams-Gray[1]

[1]Department of Clinical Neurosciences, University of Cambridge, Cambridge, UK
[2]Cambridge Clinical Trials Unit, Cambridge, UK
[3]Department of Pharmacy, Cambridge University Hospitals NHS Foundation Trust, Cambridge, UK

**Correspondence to**
Dr Julia C Greenland;
jcg69@cam.ac.uk

## ABSTRACT

**Introduction** The immune system is implicated in the aetiology and progression of Parkinson's disease (PD). Inflammation and immune activation occur both in the brain and in the periphery, and a proinflammatory cytokine profile is associated with more rapid clinical progression. Furthermore, the risk of developing PD is related to genetic variation in immune-related genes and reduced by the use of immunosuppressant medication. We are therefore conducting a 'proof of concept' trial of azathioprine, an immunosuppressant medication, to investigate whether suppressing the peripheral immune system has a disease-modifying effect in PD.

**Methods and analysis** AZA-PD is a phase II randomised placebo-controlled double-blind trial in early PD. Sixty participants, with clinical markers indicating an elevated risk of disease progression and no inflammatory or immune comorbidity, will be treated (azathioprine:placebo, 1:1) for 12 months, with a further 6-month follow-up. The primary outcome is the change in the Movement Disorder Society-Unified Parkinson's Disease Rating Scale gait/axial score in the OFF state over the 12-month treatment period. Exploratory outcomes include additional measures of motor and cognitive function, non-motor symptoms and quality of life. In addition, peripheral and central immune markers will be investigated through analysis of blood, cerebrospinal fluid and PK-11195 positron emission tomography imaging.

**Ethics and dissemination** The study was approved by the London-Westminster research ethics committee (reference 19/LO/1705) and has been accepted by the Medicines and Healthcare products Regulatory Agency (MHRA) for a clinical trials authorisation (reference CTA 12854/0248/001–0001). In addition, approval has been granted from the Administration of Radioactive Substances Advisory Committee. The results of this trial will be disseminated through publication in scientific journals and presentation at national and international conferences, and a lay summary will be available on our website.

**Trial registration numbers** ISRCTN14616801 and EudraCT- 2018-003089-14.

## Strengths and limitations of this study

► First clinical trial of a peripherally acting immuno-suppressive drug in Parkinson's disease (PD).
► Robust, randomised double-blind placebo-controlled design.
► Novel patient stratification approach with recruitment of a more rapidly progressing subgroup to optimise chance of demonstrating clinical effect.
► Detailed exploratory measures examining peripheral and central immune profile in PD to demonstrate proof of mechanism.
► As a single centre 'proof of concept' trial, sample size is limited to 60 participants.

key motor features. The core pathology in PD involves the loss of dopaminergic neurons in the substantia nigra (SN) with intracellular accumulation of alpha-synuclein aggregates (Lewy bodies). Dopamine replacement therapy can control some of the motor symptoms. However, other problems including impaired balance and cognitive dysfunction are due to more widespread neurodegenerative pathology and are consequently unresponsive to dopaminergic therapies. These symptoms progress such that by 10 years from diagnosis, around two-thirds of patients have balance and walking difficulties, and around half have dementia,[1] with a profound impact on quality of life[2 3] and care requirements.[4] There are currently no treatments to alter disease course and prevent these devastating complications, hence there is an urgent need to find effective disease-modifying therapies for PD. There is increasing evidence that the immune system plays an important role in driving neurodegeneration in PD, and we propose that targeting the immune

## INTRODUCTION

Parkinson's disease (PD) is a common neurodegenerative disorder diagnosed clinically by

**Table 1** Schedule of assessments

| | Screening visit | Imaging visit | CSF collection | Baseline visit | Monitoring visits* | Dose escalation visit | Midtreatment visit | End-of-treatment visit | Imaging visit | Follow-up visit |
|---|---|---|---|---|---|---|---|---|---|---|
| | Day −42 (max) | Approx. day −14± | Approx. day −7± | Day 0±14 | Day 14±5 and onwards | Day 28±5 | Day 182±14 | Day 365±14 | Day 410±45 | Day 547±14 |
| Informed written consent | ✓ | | | | | | | | | |
| Eligibility review | ✓ | | | | | | | | | |
| Randomisation | | | | ✓ | | | | | | |
| Vital signs | | | | ✓ | ✓ | ✓ | ✓ | ✓ | | ✓ |
| Weight in kg | ✓ | | | ✓ | ✓ | ✓ | ✓ | | | |
| Clinical assessments | | | | | | | | | | |
| Demographics | | | | ✓ | | | | | | |
| Medical history | ✓ | | | ✓ | ✓ | ✓ | | ✓ | | ✓ |
| Concomitant medication review | ✓ | | | ✓ | ✓ | ✓ | ✓ | ✓ | | ✓ |
| MDS-UPDRS | | | | ✓ | ✓ | ✓ | ✓ | ✓ | | ✓ |
| NART | | | | ✓ | | | | | | |
| ACE-III | | | | ✓ | | | ✓ | ✓ | | ✓ |
| GDS | | | | ✓ | | | ✓ | ✓ | | ✓ |
| NMSS | | | | ✓ | | | ✓ | ✓ | | ✓ |
| PDQ-39 | | | | ✓ | | | ✓ | ✓ | | ✓ |
| Adverse events review | | ✓ | ✓ | ✓ | ✓ | ✓ | ✓ | ✓ | ✓ | ✓ |
| IMP compliance check | | | | | ✓ | ✓ | ✓ | ✓ | | |
| Imaging – optional | | | | | | | | | | |
| (11C)-PK11195 PET-MRI | | ✓ | | | | | | | ✓ | |
| Blood tests | | | | | | | | | | |
| Screening bloods† | ✓ | | | | | | | | | |
| Safety monitoring bloods‡ | | | | | ✓ | ✓ | ✓ | ✓ | | ✓ |
| CRP, immunoglobulins and serum storage for cytokine measurement | | | ✓ | | ✓ | ✓ | ✓ | ✓ | | ✓ |
| Immunophenotyping | | | | ✓ | | | ✓ | ✓ | | ✓ |
| CSF – optional | | | | | | | | | | |
| Immune markers and immunophenotyping | | | ✓ | | | | | ✓ | | |

*Monitoring visits will take place at: day 14, day 42, day 56, day 70, day 98, day 252 and day 547 (as part of the routine follow-up visit). Additional monitoring visits may also be scheduled if there are patient safety concerns.
†Screening bloods include FBC, U&Es, LFTs, coagulation, TPMT, HIV, syphilis, hepB, hepC, EBV serology, VZV serology, LH and FSH (if female and reproductive age).
‡Monitoring bloods include FBC, U&Es and LFTs.
ACE-III, Addenbrooke's Cognitive Examination-III; CRP, C reactive protein; CSF, cerebrospinal fluid; EBV, Epstein-Barr virus; FBC, full blood count; FSH, Follicle-stimulating hormone; GDS, Geriatric Depression Scale; HepB, Hepatitis B; HepC, Hepatitis C; IMP, Investigational medicinal product; LFT, liver function tests; LH, Luteinising hormone; MDS-UPDRS, Movement Disorder Society-Unified Parkinson's Disease Rating Scale; NART, National adult reading test; NMSS, Non-Motor Symptom Scale; PDQ-39, Parkinson's Disease Questionnaire 39; PET, positron emission tomography; TPMT, Thiopurine methyltransferase; U&E, urea and electrolytes; VZV, Varicella zoster virus.

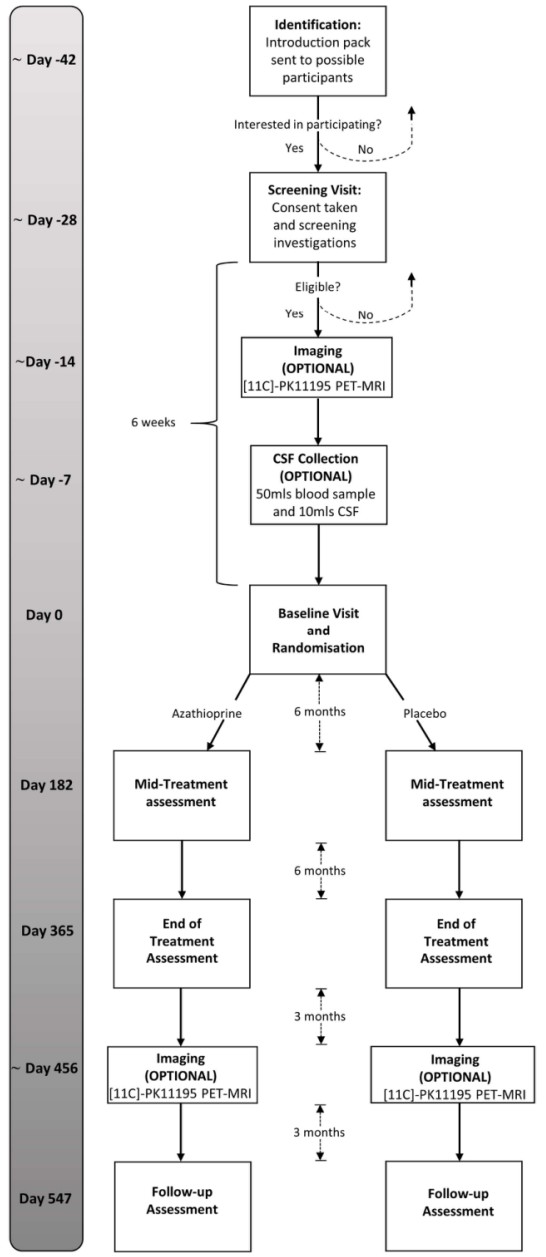

**Figure 1** Overview of trial timeline. CSF, cerebrospinal fluid; PET, positron emission tomography.

system may be an effective strategy for slowing disease progression.

The link between genetic variation in immune pathways and risk of PD is well established. Risk of developing PD is associated with polymorphisms in the human leucocyte antigen (HLA) region, which encodes proteins vital to antigen recognition and presentation.[5 6] Large-scale analyses of Genome Wide Association Study data also implicate the immune system in PD, demonstrating heritability enrichment for genes of the adaptive immune system, including those involved in lymphocyte regulation and cytokine signalling pathways.[7 8] Further evidence of an immune contribution to PD risk comes from epidemiological studies: individuals who regularly take non-steroidal anti-inflammatory drugs have reduced risk of

developing PD,[9 10] as do those on immunosuppressive therapy.[11] There is also evidence that immune activation impacts on disease progression rate. In a large incident PD cohort, a proinflammatory serum cytokine profile at baseline was associated with faster motor progression and impaired cognition over 36 months of follow-up.[12]

Activation of microglia, the inflammatory cells of the brain, has been clearly demonstrated in patients with PD both at postmortem[13–16] and using [$^{11}$C]-PK11195 positron emission tomography (PET) imaging in vivo.[17–19] These cells have a role in responding to tissue injury, regulating the cerebral microenvironment and antigen presentation.[20] It is thought that this activation is driven by toxic misfolded or post-translationally modified forms of alpha-synuclein released by degenerating cells, leading to secretion of proinflammatory and neurotoxic molecules, resulting in a cyclical process of cell damage.[21]

Abnormalities in the peripheral immune profile in PD are also well demonstrated and include alterations in both the innate and adaptive immune compartments. There is a shift towards 'classical' (inflammatory) monocytes with elevated expression of activation markers,[22] particularly in patients at higher dementia risk.[23] In the T lymphocyte compartment, several authors have reported bias towards proinflammatory CD4+ lymphocyte subsets and production of proinflammatory cytokines.[24–27] There may also be a reduction in the number and function of CD4+ T-regulatory (Treg) cells, whose role is to counter this proinflammatory response.[24 26] In addition, changes in the CD8 compartment have been reported, with increased expression of activation markers and reduced markers of age-related senescence.[28]

Importantly, T cells with specificity for epitopes of alpha-synuclein have been identified and reported to occur at higher frequency in PD than controls; furthermore, their frequency was closely associated with possession of known PD risk alleles at the HLA locus,[29] thus suggesting that alpha-synuclein may drive a peripheral adaptive immune response as well as an innate response of microglia in the brain. Elevated levels of alpha-synuclein specific antibodies are also present in the early stages of PD.[30] Peripheral immune cells may contribute to brain inflammation and neurotoxicity by trafficking into the central nervous system in PD. CD4+ and CD8+ lymphocytes have been shown to be present in increased numbers in the SN at postmortem in patients with PD,[16 25] as well as in ex vivo cerebrospinal fluid (CSF) samples.[31] The precise mechanism by which peripheral immune cells drive neuronal damage in PD is still unclear, but it has recently been demonstrated that Th17 cells from patients with PD drive cell death in autologous induced pluripotent stem cell (iPSC) derived dopaminergic midbrain neurons.[25]

Immune manipulation in animal models of PD alters disease susceptibility and severity. Using an 1-methyl-4-phenyl-1,2,3,6-tetrahydropyridine (MPTP) mouse model of PD, studies have demonstrated that a lack of CD4+ lymphocytes attenuates dopaminergic cell death,[16] as does administration of Treg cells.[32] In mice that overexpress

*Inclusion Criteria*
- be capable of giving signed informed consent
- be aged over 50 years
- be a fluent English speaker
- have a diagnosis of PD according to UKPDS Brain Bank Criteria
- have a disease duration of less than 3 years
- have a probability of poor outcome (postural instability/dementia/death) at 5 years from diagnosis ≥50% [41]
- have adequate organ and marrow function, as defined below (measured within 42 days of first dose of trial medication):
  - Haemoglobin ≥ 110 g/L
  - Platelet count ≥ 130 x $10^9$/L
  - Neutrophil count ≥ 1.5 x $10^9$/L
  - Renal function- creatinine clearance ≥50mL/min.
  - Hepatic function- ALT and bilirubin ≤2 times the institutional upper limit of normal

*Exclusion Criteria*
- Any use of immunomodulatory drugs such as azathioprine, mycophenolate, methotrexate, ciclosporin, cyclophosphamide within the 12 months prior to screening
- Any previous use of rituximab or alemtuzumab at any time
- Treatment with oral corticosteroids for greater than 2 weeks within the 12 months prior to screening, or any oral steroid use in 3 months prior to screening
- Regular use of NSAIDs including aspirin >75mg, naproxen, ibuprofen, meloxicam on more than 2 days per week
- Known inflammatory or autoimmune disease
- Chronic or latent infection
- Active infection requiring the use of parenteral antimicrobial agents within 2 months prior to the first dose of trial treatment
- Skin or solid organ malignancy within the 5 years prior to the screening assessment
- Current or previous haematological malignancy
- The inability to take or swallow oral medication
- Parkinson's Disease Dementia according to MDS PD Dementia criteria
- A positive test for HIV or Hepatitis
- TPMT deficiency
- A lack of immunity to VZV
- Negative EBV IgG
- Chronic liver disease
- Renal impairment - creatinine clearance <50mL/min
- Current or previous haematological malignancy
- Concomitant allopurinol
- Any concurrent medical or psychiatric condition or disease that is likely to interfere with the trial procedures or results, or that in the opinion of the investigator, would constitute a hazard for participating in this trial
- Receipt of live, attenuated vaccine within the 30 days prior to the screening assessment
- Women of childbearing potential. Female patients must be surgically sterile or be postmenopausal
- Male patients must be surgically sterile or must agree to use effective contraception during the period of therapy and for 6 months after the last dose of the trial treatment
- Known hypersensitivity to azathioprine or its excipient
- Received an investigational drug or used an invasive investigational medical device within 4 weeks before the screening assessment or is currently enrolled in an interventional investigational trial

**Figure 2** AZA-PD eligibility criteria. MDS, Movement Disorder Society; NSAIDs, non-steroidal anti-inflammatory drugs; PD, Parkinson's disease; UKPDS, United Kingdom Parkinson's disease society; ALT, alanine transaminase; TPMT, thiopurine methyltransferase; VZV, varicella zoster virus; EBV, Epstein-Barr virus

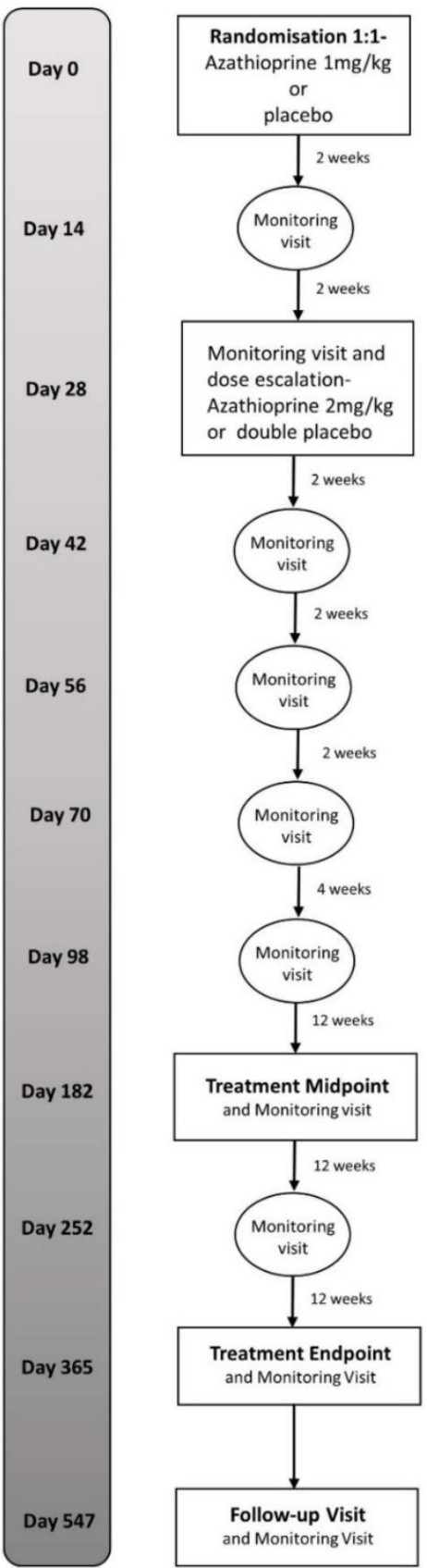

**Figure 3** Treatment monitoring schedule.

alpha-synuclein, knockout of major histocompatability class II (MHC) class II prevents both microglial activation and dopaminergic neurodegeneration.[33] Furthermore,

ciclosporin, a widely used immunosuppressant, is effective in improving motor and cognitive deficits in multiple mouse models of PD.[34]

Although animal models of PD indicate that immunomodulatory therapies may have efficacy in protecting against neurodegeneration, there is limited clinical trial data in PD to date. Phase II trials of minocycline and pioglitazone, agents that reduce microglial activation in the brain in animal models, have been negative.[35–37] An early phase trial of sargramostim, a human recombinant granulocyte-macrophage colony-stimulating factor that promotes differentiation of proinflammatory T-effector cells into Treg cells has reported a modest improvement in an exploratory outcome of motor function over 8-week treatment.[38]

We propose that direct suppression of the peripheral immune system is an alternative, highly relevant therapeutic strategy that has not been tested in clinical trials to date. Azathioprine is an immunosuppressant drug widely used in clinical practice for a range of immune-related conditions. It is a purine analogue that inhibits nucleic acid synthesis, hence reducing proliferation of lymphocytes involved in targeting and amplification of the immune response. It affects both the cell-mediated and antibody-mediated responses through reducing T and B lymphocyte proliferation.[39] It was selected over other immunosuppressants because of its established efficacy in a range of clinical conditions, including central nervous system disorders such as multiple sclerosis,[40] and its acceptable safety profile with recognised protocols for toxicity monitoring. Furthermore, it is generally well tolerated in the elderly and is a once-daily preparation for ease of administration.

## METHODS AND ANALYSIS
### Overview
AZA-PD is a randomised, double-blind, placebo-controlled trial of azathioprine in early PD that aims to provide 'proof of concept' that a peripherally acting immunosuppressive drug can slow clinical disease progression. The trial will investigate whether the drug has an effect on disease course over 12 months of treatment and whether this is maintained over 6 months of subsequent follow-up. Sixty participants will be recruited and randomised 1:1 to receive active treatment or placebo. Clinical assessments will be performed at baseline, 6 months, 12 months and 18 months (6 months postcompletion of treatment), with rigorous safety monitoring. In addition, the trial aims to demonstrate 'proof of mechanism' by evaluating the impact of azathioprine on blood, CSF and neuroimaging parameters of immune activation in the trial population and determining the relationship between these parameters and clinical measures of disease progression.

The trial timeline is summarised in figure 1 .

Although AZA-PD is open to recruitment, given the current COVID-19 pandemic, recruitment has not commenced due to safety concerns. The Trial Steering

Committee (TSC) and Data and Safety Monitoring Board (DSMB) will decide on an appropriate date to begin recruitment in due course, and protocol amendments to maximise patient safety will be submitted to the appropriate bodies when the best course of action has been determined. Our current aim is to start recruitment in March 2021, closing to recruitment in March 2022, with last patient last visit in November 2023, although this may be subject to change depending on the COVID-19 pandemic.

## Patient and public involvement (PPI)

Patients with PD and their partners and carers who attend our research clinic at the John van Geest Centre for Brain Repair (VGB), University of Cambridge, gave input into the protocol design. A PPI advisory panel of four patients/carers reviewed the protocol and provided specific feedback, leading to the addition of optional components. The PPI panel also reviewed our participant information sheet for clarity.

## Participant identification

Participants will be recruited from a single site in Cambridge, UK. Potential participants will be identified from the PD Research clinic database at the VGB. These individuals have undergone detailed clinical phenotyping, and information on demographics, comorbidities and medication is available. They have consented to be contacted about other research studies. Potential participants will be preselected by cross-referencing existing data with the inclusion/exclusion criteria outlined in figure 2. A key component of this process involves the calculation of a prognostic risk score, based on a model we have previously developed and validated, using age, Movement Disorder Society-Unified Parkinson's Disease Rating Scale (MDS-UPDRS) axial score and semantic fluency to estimate risk of a poor prognosis (dementia, postural instability or death) within 5 years.[41] Only patients with a risk greater than 50%, based on prior assessment at the research clinic, will be invited to take part. Approximately 40% of those on the database fall within this group. This strategy has been adopted to maximise the probability of demonstrating significant slowing of clinical progression with azathioprine treatment.

Potential participants will be sent a copy of the participant information sheet and telephoned after 2 weeks to determine whether they are interested in participating. If so, they will attend a screening visit, where written informed consent will be obtained before confirming eligibility.

## Eligibility criteria

A potential participant will be deemed eligible for recruitment into AZA-PD if they meet the inclusion and exclusion criteria listed in figure 2. A review of medical history and blood tests will be used to determine eligibility.

## Outcome measures

The primary outcome measure is change in MDS-UPDRS gait/axial score in the OFF state over the 12-month treatment period. This is a clinical measure that has been shown to be the most sensitive measure of motor progression in PD, is relatively resistant to dopaminergic therapy and has an important impact on quality of life.[42] This score is a sum of the points from the following sections of MDS-UPDRS part III:

► 3.1 – speech.
► 3.2 – facial expression.
► 3.9 – rising from a chair.
► 3.10 – gait.
► 3.12 – postural stability.
► 3.13 – posture.
► 3.14 – body bradykinesia.

Other outcome measures are exploratory and include:

► Change in MDS-UPDRS gait/axial score in OFF state at 18 months.
► Change in total MDS-UPDRS in OFF state at 12 and 18 months.
► Change in electromagnetic sensor (EMS) measurements while performing MDS-UPDRS tremor and bradykinesia assessments at 12 and 18 months.
► Proportion of patients developing postural instability (Hoehn and Yahr stage 3 or greater) at 12 and 18 months.
► Change in global cognition (Addenbrooke's Cognitive Examination-III (ACE-III)) at 12 and 18 months.
► Change in patient-reported outcome measure of quality of life (Parkinson's Disease Questionnaire 39 (PDQ-39)) at 12 and 18 months.
► Change in Non-Motor Symptom Scale (NMSS) at 12 and 18 months.
► Change in dose of symptomatic dopaminergic therapy (Levodopa Equivalent Daily Dose (LEDD)) at 12 and 18 months.
► The safety and tolerability of azathioprine assessed by the number of adverse events (AEs) recorded during the 12-month treatment period.
► Change in $[^{11}C]$-PK11195 PET non-dissociable binding potential ($BP_{ND}$) in subcortical and cortical regions of interest at 12 months.
► Change in total lymphocyte count at 6, 12 and 18 months.
► Change in serum immunoglobulin levels at 6, 12 and 18 months.
► Change in levels of serum and CSF cytokine levels at 12 and 18 months.
► Change in lymphocyte subsets in blood and CSF at 12 and 18 months.

## Sample size calculation

The treatment effect size is unknown and therefore cannot be used to inform sample size calculations. A sample size of 60 has been selected pragmatically based on feasibility of recruitment.

However, longitudinal clinical data from the Incidence of Cognitive Impairment in Cohorts with Longitudinal Evaluation-PD (ICICLE-PD) cohort study provides some idea of an anticipated effect size for the primary outcome measure. ICICLE-PD patients were stratified by cytokine profile. Those with a 'proinflammatory' profile (n=32) had a more rapid symptom progression, with mean (SD) annualised change in MDS-UPDRS gait/axial score of 1.95 (1.92). In the subgroup with an 'anti-inflammatory' cytokine profile (n=26), mean (SD) annualised change in MDS-UPDRS gait/axial score was 0.72 (1.40).[12] The corresponding between-group difference of 1.2 points is equivalent to a standardised effect size (Cohen's d) of 0.73. The magnitude of this effect, which equates to a 4% change on the 28-point gait/axial MDS-UPDRS subscale, would be clinically significant. For comparison, the estimated minimum clinically important change on the full 132-point MDS-UPDRS motor scale is $\approx$2% (2.5 points).[43] Furthermore, the axial-gait items of the MDS-UPDRS are those with the greatest impact on quality of life.[42]

As this is an early-phase proof of concept trial, it is important to maximise the chances of continuing development if the treatment is genuinely effective, and thus a significance level of 25% under a one-sided test will be used. If the treatment effect is a 2% change (0.37 standardised effect), the design has 78% power, and for a 4% change (0.73 standardised effect), the design has 99% power.

## Trial procedures
### Clinical
Clinical measures assessing both motor and non-motor components of PD will be performed at baseline, midtreatment, end of treatment and after 6-month further follow-up (see table 1). Throughout the course of the trial, participants will continue to take their PD medication as prescribed by their treating physician, and dose adjustments are permitted. However, some assessments will be conducted 'OFF' medication.

The MDS-UPDRS is widely used to quantify PD severity and includes questionnaires assessing the non-motor and motor aspects of the disease, a motor examination performed by a clinician and an assessment of motor complications (dyskinesias and fluctuations).[44] The MDS-UPDRS part III will be assessed in the OFF state, in the absence of regular dopaminergic medication, with participants being asked to not take their levodopa in the 8 hours prior to the assessment or their long-acting agents (eg, ropinirole, pramipexole and rasagiline) in the preceding 36 hours. The aim of this is to expose underlying disease severity and avoid confounding effects from variability in medication doses or timing. This examination will be filmed to enable subsequent rating by an independent assessor to check inter-rater reliability. Our primary outcome measure is derived from the MDS-UPDRS: the gait/axial subscore, as previously discussed. Two sections of the MDS-UPDRS part III (tremor and bradykinesia) will be repeated while the participant is

wearing an EMS (Polhemus Inc) on the index finger and thumb, which will give an objective measure of the participant's movements. Motor stage will also be evaluated using the Hoehn and Yahr scale, a five-point scale used to capture the stages of progression of PD, with stage 3 representing the development of postural instability.[45]

Cognition will be assessed using the ACE-III. This provides a global measure of cognition as well as subscores in five domains; attention, memory, fluency, language and visuospatial function.[46] Other non-motor aspects of PD will be evaluated using the short form 15-item Geriatric Depression Scale, a questionnaire assessing depressive symptoms filled in by the participant,[47] and the PD NMSS, completed by the trial assessor.[48] Finally, we will use the PDQ-39, a self-rated questionnaire measuring PD-related quality of life.[49]

Dopaminergic medication requirement will be monitored throughout the trial and standardised by calculating LEDD, which allows quantification of different doses and types of Parkinson's medication on a single scale.[50]

## PK-11195 PET imaging
$[^{11}C]$-PK11195 PET will be used to measure activated microglia in the brain.[17 18] Scanning will be conducted at the Wolfson Brain Imaging Centre (WBIC) on a GE SIGNA PET/MRI scanner, with the radiotracer produced at the WBIC Radiopharmaceutical Chemistry laboratory. MRI will be used for colocalisation. Five hundred megabecquerel of the $[^{11}C]$-PK11195 radiotracer will be injected via a peripheral venous cannula over 30 s, and PET emission data will be acquired for 75 min postinjection in 55 time frames. Following image reconstruction and attenuation correction, specific tracer binding will be analysed with the simplified reference tissue model[51] to quantify binding potential relative to a non-displaceable compartment ($BP_{ND}$). The reference region will be estimated with supervised cluster analysis for $[^{11}C]$PK11195 from existing scans in healthy controls acquired on the same scanner. BPND will be compared pretreatment and post-treatment using a region of interest approach.

Given that some participants may have difficulty tolerating prolonged PET imaging, this will be optional. It will be performed between screening and baseline and repeated within 3 months following the end of treatment.

## Biosample collection and processing
Fourteen millilitres of blood will be collected in serum tubes at baseline, midtreatment, end-of-treatment and follow-up visits for analysis of inflammatory cytokines, C reactive protein and immunoglobulins. Tubes will be centrifuged at 2000 RPM (600G) for 15 min following 15 min clotting time for extraction of serum. Aliquots will be stored at −80°C for subsequent batch analysis using ELISA and electrochemiluminescence assays.

At baseline, end of treatment and follow-up visits, an additional 27 mL of blood will be collected in lithium heparin tubes for separation of peripheral blood mononuclear cells (PBMCs) for immunophenotyping. A

concurrent full blood count (FBC) will be performed from an EDTA sample (2.6 mL).

CSF will be collected via lumbar puncture before the baseline visit and at the treatment endpoint. This is an optional component of the study in order to ensure that its inclusion does not limit recruitment. CSF will be spun at 400G for 10 min for extraction of immune cells for contemporaneous immunophenotyping alongside PBMC analysis. Supernatants will be stored at −80°C for later batch analysis of relevant immune and protein markers. Immunophenotyping will be performed for subsets of T cells, B cells and monocytes using flow cytometry, run within 24 hours of sample collection.

### Treatment allocation, blinding and safety monitoring

Participants will be randomised 1:1 to receive azathioprine or placebo using Sealed Envelope, an online randomisation system. Clinical assessors and participants will be blinded to treatment allocation. Balanced assignment of each treatment will be achieved using permuted block randomisation, which will be stratified for: age ≤71 versus >71, and MDS-UPDRS-III ≤30 versus >30.

Treatment will be commenced at a dose of 1 mg/kg, based on 25 mg tablets of IMP (azathioprine/placebo). In addition to the visits shown in figure 1, treatment monitoring visits will be conducted to screen for potential complications associated with azathioprine. These will include blood tests to screen for myelosuppression, liver or renal dysfunction, AEs reporting and assessment of treatment compliance (review of patient-completed dosing diary and counting of the investigational medicinal product (IMP) at regular intervals). Initially, monitoring visits will occur 2 weekly, and after 4 weeks, the azathioprine dose will be increased to 2 mg/kg (assuming blood tests and clinical assessments are satisfactory), the standard therapeutic dose used in clinical practice. There will be a matched doubling of the placebo dose to maintain blinding. Once the participant is stable on their dose, treatment monitoring will be carried out less frequently (see monitoring protocol, figure 3).

Given that azathioprine will produce changes in FBC parameters, the blinded trial team conducting patient assessments and laboratory analysis will not have access to monitoring blood results throughout the duration of the trial. The blood tests will be reviewed by a separate unblinded team of clinicians, who will make decisions on dose changes when necessary. If a dose reduction is required, the participant will have an additional three monitoring visits at 2-weekly intervals to ensure stability of blood tests. Dose reductions and, where necessary, withdrawal of treatment will be carried out based on predefined clinical and laboratory criteria to ensure the safety of participants, including the development of significant myelosuppression, intolerable gastrointestinal side effects and hypersensitivity reactions. Participants who have been withdrawn from treatment will be encouraged to continue to attend the remainder of the trial assessments.

To ensure blinding is maintained among clinical assessors and participants, dose adjustments and treatment withdrawals will be also made for an equal number of participants in the placebo arm, with additional monitoring visits. Matched pairs of placebo and azathioprine-treated participants will be generated to facilitate this, and all dose adjustment decisions will be made by the unblinded team.

Emergency unblinding will be carried out in the event of a valid medical or safety reason, where the clinical care of the participant will be facilitated by the knowledge of whether they have been taking azathioprine, as decided by the treating clinician. It will be executed using Sealed Envelope, and where possible, the trial team will remain blinded.

Following the end of the trial, and for participants who withdraw early, we will offer continuing follow-up through our research clinic at the VGB.

### Trial monitoring and oversight

Safety monitoring will be overseen by a DSMB who will have access to interim recruitment and safety data. The DSMB will report to the TSC should it become clear that one treatment allocation is either indicated or contraindicated, or apparent that no clear outcome can be obtained from the trial. The TSC, who are independent from the sponsor, will provide overall supervision of the trial and ensure that it meets appropriate standards. These groups include clinicians with experience in PD or immunosuppression and independent statisticians, and the TSC includes a lay member.

AZA-PD is jointly sponsored by Cambridge University Hospitals Natioanl Health Service Foundation Trust and the University of Cambridge. The sponsor will review all trial documentation, including any proposed amendments, prior to submission to the relevant regulatory bodies, which can only be completed once the sponsor has approved the changes. Changes will then be communicated to participants, the DMSB, TSC and trial registries.

Adherence to the protocol and regulatory requirements will be reviewed by a Clinical Trials Monitor, assigned by the sponsor. The first monitoring visit will occur within 10 days of the first randomisation, with frequency thereafter determined by a risk assessment, which will be reviewed and adjusted as necessary throughout the course of the trial.

## DATA ANALYSIS

Trial data will be transferred from paper case report forms (CRFs) to the electronic trial database, where it will be anonymised, but with preserved linkage records. Patient-identifiable data (PID) will be stored on a password-protected database within the Secure Data Hosting Service (SDHS) hosted by the University of Cambridge, with access granted only to relevant members of the trial team. PID will be kept for 5 years following the end of the trial, as per regulatory requirements. Participant consent

will be specifically sought for data/sample sharing with our collaborators and use of remaining biological samples in future ethically approved research.

Data will be analysed on an 'intention to treat' basis, with further 'per protocol' analysis in participants with at least 80% compliance with trial medication. All endpoints will be summarised and broken down by treatment group and time point, where relevant. Mean, median, SD and minimum/maximum will be used for continuous endpoints and frequency tables for categorical or binary endpoints. Equivalent box and whisker plots or stacked bar charts will be produced for continuous and categorical endpoints, respectively.

The primary analysis will estimate the difference between treatment groups in terms of the primary endpoint. An analysis of covariance (ANCOVA) model will be fitted adjusting for baseline MDS-UPDRS gait/axial score, gender, LEDD and age. Treatment effect estimates, SEs, CIs (95% and 40% levels) and one-sided p values will be provided. A one-sided p value less than 25% will be regarded as statistically significant. Similar comparative analyses will be produced for other time points of the MDS-UPDRS gait/axial score and exploratory endpoints, using ANCOVA for continuous endpoints or logistic regression for categorical or binary endpoints.

Longitudinal data will use a mixed effect model repeated measurements (MMRM) analysis to include an unstructured patient-level random effect for nominal visit, visit and visit–treatment interaction fixed effects at visits post-baseline, with adjustment for baseline covariates. To assess the slope of change over time, the longitudinal data will be analysed using a similar MMRM but with a fixed effect of time from randomisation as a continuous, rather than nominal covariate, with a treatment–time interaction to compare treatment groups and patient-level random effect for slope, with adjustment for baseline covariates.

## ETHICS AND DISSEMINATION

This study was approved by the London-Westminster research ethics committee (reference 19/LO/1705) and has been accepted by the MHRA for a clinical trials authorisation (reference CTA 12854/0248/001–0001). In addition, approval has been granted from the Administration of Radioactive Substances Advisory Committee.

We will feedback trial results to participants and our wider cohort of research participants via our annual PD Open Day and newsletter. A lay summary of the results will be available on our website. The results will also be disseminated through publication in scientific journals and presentation at national and international conferences.

AZA-PD has been accepted onto the National Institute for Health Research Clinical Research Network portfolio.

**Acknowledgements** This research was supported by the National Institute for Health Research (NIHR) Cambridge Biomedical Research Centre and the Cambridge Clinical Trials Unit.

**Contributors** CW-G is the CI of this trial. JG is subinvestigator of this trial. CW-G and JG: study design and writing the protocol. SK, EC and AC: critical review of the protocol. SB: statistical analysis plan.

**Funding** This trial is funded by Cambridge Centre for Parkinson-Plus (grant no: RG95450) and the Cure Parkinson's Trust (grant ref CW011) and supported by the NIHR Cambridge Biomedical Research Centre (grant ref no 146281). CW-G is supported by a Research Councils UK/UK Research and Innovation (RCUK/UKRI) Research Innovation Fellowship awarded by the Medical Research Council (MR/R007446/1). This study is jointly sponsored by the University of Cambridge and Cambridge University Hospitals NHS foundation Trust.

**Disclaimer** The funders have no role in study design; collection, management, analysis, and interpretation of data; writing of the report; or the decision to submit the report for publication.

**Competing interests** None declared.

**Patient and public involvement** Patients and/or the public were involved in the design, or conduct, or reporting, or dissemination plans of this research. Refer to the Methods section for further details.

**Patient consent for publication** Not required.

**Provenance and peer review** Not commissioned; externally peer reviewed.

**ORCID iD**
Julia C Greenland http://orcid.org/0000-0001-9267-0586

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
