## [Reviewer comments · BMJ Open]

ARTICLE DETAILS

TITLE (PROVISIONAL)	Azathioprine immunosuppression and disease modification in Parkinson's disease (AZA-PD): a randomised double-blind placebo-controlled phase II trial protocol
AUTHORS	Greenland, Julia; Cutting, Emma; Kadyan, Sonakshi; Bond, Simon; Chhabra, Anita; Williams-Gray, Caroline

VERSION 1 – REVIEW

REVIEWER	Philippe Desmarais Centre de Recherche du Centre Hospitalier de l'Université de Montréal - Axe Neurosciences, Montréal, Québec, Canada
REVIEW RETURNED	15-Jun-2020

GENERAL COMMENTS	Dr Greenland and colleagues are reporting the protocol of their randomised double-blind placebo-controlled study of azathioprine as a potential disease modifying drug for Parkinson's disease. The protocol is well written, clear, and detailed. I have no questions or corrections to suggest. Thank you,
---

REVIEWER	Professor Tom Foltynie UCL Queen Square Institute of Neurology, UK
REVIEW RETURNED	01-Jul-2020

GENERAL COMMENTS	This is a well written manuscript that fulfils its objectives. I have only 2 minor comments; 1)- Clarify the process for performing OFF medication assessments, i.e. the practically defined OFF medication state. 2)- Page 21. Clarify that withdrawals will be made for "an equal number" of participants in the placebo arm.
--

REVIEWER	Angus Macleod University of Aberdeen
REVIEW RETURNED	06-Jul-2020

GENERAL COMMENTS	This is well-written trial protocol describing a small randomised controlled trial of azathioprine for disease modification in early Parkinson's disease. The aim is to provide proof of concept that modification of the peripheral immune system can modify disease progression. The use of a low-cost drug with known safety profile is a clear advantage. The rationale is clearly stated and the methods have been carefully considered to perform a RCT robustly.
--

	My main concern about this trial is the short duration and small sample size. Given the anticipated effect size (1.2 points per year in the primary outcome) is small, and I am unsure if this is a clinically meaningful change, a longer trial duration would have a higher chance of finding a statistically and clinically significant result. Similarly, the sample size is small with power calculations based on a high alpha value to give reasonable power estimates. An interesting aspect of the methodology is the use of a prognostic model to select patients with higher predicted rates of disease progression. This is clearly an advantage in situations where an intervention is expensive (and resources are limited) or where a treatment carries high risk of serious adverse effects, but I am unclear how useful this will be in the context of this trial. Will this limit sample size and power compared to recruiting all-comers? It would also be helpful if in the paragraph "Participant identification" you specified the proportion of otherwise potential subjects (i.e. recently diagnosed PD in your clinic) you expect will have a risk >50% of poor outcome at 5 years. Please clarify the timing of the prognostic model assessment (at diagnosis or at trial recruitment). If later than diagnosis, please consider whether the prognostic model has been validated at this time point. I also wonder whether a better approach would be to recruit patients with a pro-inflammatory profile instead of those with faster predicted disease progression? Do the authors have any data on the proportion of those with a predicted 50% poor outcome have a pro-inflammatory profile?
--	---

VERSION 1 – AUTHOR RESPONSE

Reviewer(s)' Comments to Author:

Reviewer: 1

Reviewer Name: Philippe Desmarais

Institution and Country: Centre de Recherche du Centre Hospitalier de l'Université de Montréal - Axe Neurosciences, Montréal, Québec, Canada

Competing interests: None declared

Please leave your comments for the authors below

Dr Greenland and colleagues are reporting the protocol of their randomised double-blind placebo-controlled study of azathioprine as a potential disease modifying drug for Parkinson's disease. The protocol is well written, clear, and detailed.

I have no questions or corrections to suggest.

Thank you,

Philippe Desmarais, MD, FRCPC, MHSc

Internist-Geriatrician

Cognitive Disorders Clinic

Centre Hospitalier de l'Université de Montréal

Reviewer: 2

Reviewer Name: Professor Tom Foltynie

Institution and Country: UCL Queen Square Institute of Neurology, UK

Competing interests: None declared.

Please leave your comments for the authors below

This is a well written manuscript that fulfils its objectives. I have only 2 minor comments;

1)- Clarify the process for performing OFF medication assessments, i.e. the practically defined OFF medication state.

Thank you- we have clarified this in the Trial procedures section.

2)- Page 21. Clarify that withdrawals will be made for "an equal number" of participants in the placebo arm.

Thank you- we have clarified this in the treatment allocation, blinding and safety monitoring section.

Reviewer: 3

Reviewer Name: Angus Macleod

Institution and Country: University of Aberdeen

Competing interests: None declared

Please leave your comments for the authors below

This is well-written trial protocol describing a small randomised controlled trial of azathioprine for disease modification in early Parkinson's disease. The aim is to provide proof of concept that modification of the peripheral immune system can modify disease progression. The use of a low-cost drug with known safety profile is a clear advantage. The rationale is clearly stated and the methods have been carefully considered to perform a RCT robustly.

My main concern about this trial is the short duration and small sample size. Given the anticipated effect size (1.2 points per year in the primary outcome) is small, and I am unsure if this is a clinically meaningful change, a longer trial duration would have a higher chance of finding a statistically and clinically significant result. Similarly, the sample size is small with power calculations based on a high alpha value to give reasonable power estimates.

We agree that sample size and treatment duration are potential limitations, but given that this is an early phase trial and we have no robust data to allow formal sample size calculations, we have chosen these parameters pragmatically based on feasibility of recruitment and available funding. The main aim of this early trial to provide proof of concept, and to generate sufficient data to inform the design of subsequent larger trials. As outlined in the manuscript, our current trial design allows 78% power for detecting a 2% effect and 99% power for detecting a 4% effect. We think that an anticipated effect size of between 2 and 4% is not unreasonable, based on our prior analysis comparing subgroups of patients from the ICICLE-PD study with 'pro-inflammatory' and 'anti-inflammatory' cytokine profiles, which showed a between-group difference of 4% in annualised change in the MDS-UPDRS gait-axial score. We hope that such an effect would be clinically relevant, as it is comparable to the estimated minimum clinically important change on the UPDRS-III reported in the literature.

An interesting aspect of the methodology is the use of a prognostic model to select patients with higher predicted rates of disease progression. This is clearly an advantage in situations where an intervention is expensive (and resources are limited) or where a treatment carries high risk of serious adverse effects, but I am unclear how useful this will be in the context of this trial. Will this limit sample size and power compared to recruiting all-comers? It would also be helpful if in the paragraph "Participant identification" you specified the proportion of otherwise potential subjects (i.e. recently

diagnosed PD in your clinic) you expect will have a risk >50% of poor outcome at 5 years. Please clarify the timing of the prognostic model assessment (at diagnosis or at trial recruitment). If later than diagnosis, please consider whether the prognostic model has been validated at this time point. I also wonder whether a better approach would be to recruit patients with a pro-inflammatory profile instead of those with faster predicted disease progression? Do the authors have any data on the proportion of those with a predicted 50% poor outcome have a pro-inflammatory profile?

Thank you for your thoughts. Our aim of selecting a group who are predicted to progress more quickly is that we will be more likely to see a treatment effect. Recruitment for most trials is biased towards younger patients with relatively benign disease – in such individuals, little progression would be expected over the course of 12 months. So if we recruited ‘all comers’, we anticipate that we would need a much larger sample size and longer treatment duration to demonstrate a treatment effect. This would make the trial economically unfeasible, and is less pragmatic for this type of ‘proof of concept’ study.

On review of our existing research clinic database, around 40% of our participants are in the ‘high risk’ prognostic group- we have added this to the participant identification section, as well as clarifying that the prognostic score will be based on their prior assessment in the PD research clinic, not calculated at the point of trial recruitment. The majority of our patients are assessed in our research clinic within a few months of diagnosis. We anticipate that the average disease duration at the point of calculation of the prognostic score will be similar to that in the CamPaIGN cohort (mean 0.3, SD 0.4 years), which was used for validation of the model.

Using the pro-inflammatory cytokine profile to stratify patients for trial entry is certainly an interesting idea, but we were not able to adopt this approach because we do not yet have a validated prognostic model based on cytokine levels. We hope to be able to develop such a strategy for use in future trials, but this will likely be based around more detailed immunophenotyping measures and will require significant additional resources for participant screening. We don’t currently have any data on the proportion of people who are high risk of poor outcome based on the risk calculator who have a pro-inflammatory profile, but we will be able to explore this through the data collected during this trial.

VERSION 2 – REVIEW

REVIEWER	Professor Tom Foltynie UCL Institute of Neurology
REVIEW RETURNED	02-Sep-2020

GENERAL COMMENTS	The authors have addressed my comments. There are however still numerous ERROR! REFERENCE SOURCE NOT FOUND throughout the revised manuscript- presumably these are embedded links to other documents/ websites. these will need to be removed/ edited during the proofing process. I have no other concerns about this manuscript which is now suitable for publication.
--

REVIEWER	Angus Macleod University of Aberdeen, UK
REVIEW RETURNED	22-Sep-2020

GENERAL COMMENTS	I have no further comments
----------------------------